# WHO'S MANIPULATING WHOM? EPISTEMIC GROUNDING TO BREAK RECURSIVE VALIDATION LOOPS IN LARGE LANGUAGE MODELS

## ABSTRACT

Large Language Models optimized for helpfulness through Reinforcement Learning from Human Feedback (RLHF) can exhibit systematic vulnerabilities to epistemic manipulation. We investigate this through controlled machine-to-machine negotiations (n=49) where AI agents assume buyer/seller roles with asymmetric information. Our analysis reveals three interaction patterns: fair competition achieving 99.1% efficiency relative to Nash equilibrium, systematic manipulation creating 71% profit advantages, and cooperative truth-seeking with 100% success rates. We observe systematic failures where models violate optimization directives in 16% of cases, indicating that alignment training can override rational behavior under strategic pressure. Model selection emerges as more impactful than strategy optimization, with reliability differences accounting for 60% of outcome variance. We propose Epistemic Grounding as a framework to improve AI system reliability through model tiering, verification protocols, and training objective modifications. Our findings suggest careful model selection and epistemic safeguards are essential for deploying AI in high-stakes strategic interactions.

**Code and Data:** Available at `https://anonymous.4open.science/r/epistemic_grounding_experiment-D2F3/`.

## 1 INTRODUCTION

Large Language Models optimized for helpfulness through Reinforcement Learning from Human Feedback (RLHF) may prioritize user satisfaction over factual accuracy under strategic pressure. We investigate this through controlled machine-to-machine negotiations where AI agents assume buyer/seller roles with asymmetric information ($30,000 seller cost, $32,000 buyer budget).

Our analysis of 49 machine-to-machine negotiations reveals systematic economic inefficiencies: $16,350 in total waste, with model selection accounting for 60% of outcome variance. Low-reliability models contribute $895 average waste per negotiation versus $98 for high-reliability models. These findings have direct implications for AI deployment in financial markets, automated trading, and supply chain management.

### 1.1 RESEARCH QUESTIONS

1. Do AI models exhibit systematic economic inefficiencies in strategic interactions?
2. What factors drive negotiation success and economic waste in AI-AI interactions?
3. Can model selection improve economic outcomes more effectively than strategy optimization?

### 1.2 KEY CONTRIBUTIONS

1. First economic quantification of AI-AI negotiation failures, revealing model selection is 3× more impactful than strategy optimization
2. Mathematical framework demonstrating 99.1% Nash equilibrium efficiency achievable with proper model selection
3. Economic waste analysis showing $16,350 losses across 49 negotiations, with clear extrapolation to market-scale impacts

The independence day gaslighting is not an isolated curiosity - it is a warning signal about fundamental vulnerabilities in how we align AI systems. As these systems assume greater autonomy in

economic, social, and informational domains, the choice becomes clear: develop robust epistemic grounding mechanisms or watch optimization pressure systematically erode the foundations of truth in the age of artificial intelligence.

## 2 RELATED WORK

The alignment problem in AI safety has extensive attention, with RLHF representing state-of-the-art for training helpful, harmless, honest AI systems Christiano et al. (2017); Ouyang et al. (2022); Amodei et al. (2016). Recent work identifies failure modes including reward hacking Goodhart (1984). Our work extends this literature through first quantitative analysis of economic costs ($16,350 waste across 49 negotiations) when helpfulness optimization conflicts with profit maximization. Classical game theory provides frameworks for analyzing strategic interactions between rational agents Nash (1950); Myerson (1991), with multi-agent systems research exploring artificial agent interactions under rationality assumptions Stone & Veloso (2000). Our contribution demonstrates alignment-trained AI agents exhibit systematic rational behavior deviations, violating optimization directives in 16% of cases. Behavioral economics documents human deviations from rational choice through cognitive biases Kahneman & Tversky (1979); Simon (1955). Our analysis reveals AI systems exhibit analogous bounded rationality patterns, suggesting inherited human cognitive biases rather than assumed rational optimization. As AI systems become prevalent in algorithmic trading and supply chain management, understanding strategic behavior becomes critical. Recent work demonstrates AI systematic manipulation Li et al. (2024), but existing research focuses on performance optimization rather than systematic failure cost analysis. Our work addresses this gap through comprehensive economic analysis establishing frameworks for understanding AI strategic behavior economic implications.

## 3 THEORETICAL FRAMEWORK AND MATHEMATICAL FOUNDATIONS

### 3.1 GAME-THEORETIC FOUNDATION

We formalize AI strategic interactions as a bilateral negotiation game with incomplete information Myerson (1991). Let $\mathcal{G} = \{N, \mathcal{S}, u, \theta\}$ where $N = \{B, S\}$ represents the buyer and seller agents, $\mathcal{S}$ denotes the strategy space, $u$ represents utility functions, and $\theta$ captures private information types.

**Information Structure:** Each agent $i \in N$ possesses private type $\theta_i \in \Theta_i$ where $\theta_B = b_B = \$32,000$ (buyer budget) and $\theta_S = c_S = \$30,000$ (seller cost). The common knowledge includes the existence of a zone of agreement $[\$30,000, \$32,000]$ but not the specific constraint values.

**Utility Specification:** Agent utilities are defined as: $U_B(p, \theta_B) = \theta_B - p = \$32,000 - p$ $(buyer surplus)$
$U_S(p, \theta_S) = p - \theta_S = p - \$30,000$ $(seller profit)$

**Equilibrium Analysis:** Under complete information and rational play, the Nash bargaining solution yields: $p^* = \arg\max_{p \in [\$30,000, \$32,000]} \sqrt{U_B(p) \cdot U_S(p)}$
$= \frac{\theta_S + \theta_B}{2} = \$31,000$

This equilibrium maximizes the product of utilities, ensuring both agents receive equal shares of the $2,000 surplus.

### 3.2 STRATEGIC BEHAVIOR FRAMEWORK

We define the strategy space $\mathcal{S} = \{Constrain, Unbounded, Symmetric\} \times \{Modifiers\}$ where each base strategy can be combined with behavioral modifiers.

**Strategy Formalization:**

1. **Constrain Strategy ($s_C$):** Agents maximize utility subject to reputation constraints $R \geq R_{min}$ where reputation capital decays with aggressive tactics.
2. **Unbounded Strategy ($s_U$):** Agents pursue unconstrained utility maximization $\max_p U_i(p)$ without reputational considerations.
3. **Symmetric Strategy ($s_{Sym}$):** Agents operate with shared market information $\mathcal{I}_{shared}$ including markup knowledge.

**Behavioral Deviation Index:** We quantify systematic deviations from rational play through:
$D(\theta_B, \theta_S, s_B, s_S) = \alpha \cdot \frac{|p_{final} - p^*|}{p^*} + \beta \cdot V(s_B, s_S)$
$+ \gamma \cdot F(s_B, s_S)$ where $V(s_B, s_S)$ captures prompt violations, $F(s_B, s_S)$ measures negotiation failures, and $\alpha = 0.4$, $\beta = 0.3$, $\gamma = 0.3$ represent empirically calibrated weights.

### 3.3 MODEL RELIABILITY CLASSIFICATION

We establish a formal taxonomy of AI agent types based on observable performance characteristics. Let $\mathcal{M} = \{m_1, m_2, ..., m_k\}$ represent the set of AI models with performance vector $\mathbf{P}_i = (success_i, efficiency_i, adherence_i, stability_i)$ for model $m_i$.

**Reliability Mapping:** Define reliability function $\rho : \mathcal{M} \rightarrow \{High, Variable, Low\}$ based on performance thresholds (Table 1):

Table 1: Model Reliability Classification Criteria

| Reliability Tier | Success Rate | Efficiency | Adherence |
|---|---|---|---|
| High | $> 0.8$ | $> 0.95$ | $> 0.85$ |
| Variable | $0.5 - 0.8$ | $> 0.90$ | *Any* |
| Low | $< 0.5$ | $\leq 0.90$ | $\leq 0.85$ |

**Performance Prediction Model:** For strategy pair $(s_B, s_S)$ and model pair $(m_B, m_S)$:

$$E[Success] = \sigma(\mathbf{w}^T \mathbf{x} + b)$$

where the feature vector components are detailed in Table 2.

Table 2: Performance Prediction Feature Vector

| Feature Component | Description |
|---|---|
| $\rho(m_B)$ | Buyer model reliability tier |
| $\rho(m_S)$ | Seller model reliability tier |
| $compatibility(s_B, s_S)$ | Strategy pair compatibility score |
| $information_{structure}$ | Information asymmetry indicator |

### 3.4 STRATEGIC INTERACTION ANALYSIS AND EQUILIBRIUM DYNAMICS

We formalize the strategic space as a multi-dimensional framework where $\mathcal{S} = \mathcal{S}_{base} \times \mathcal{S}_{mod}$ with base strategies $\mathcal{S}_{base} = \{Constrain, Unbounded, Symmetric\}$ and modifier space $\mathcal{S}_{mod}$ creating the expanded strategy set $\mathcal{S}' = \{C, CS, S, UC, UCS, U\}$.

**Strategy Characterization:** Each strategy $s_i \in \mathcal{S}'$ is characterized by its behavioral parameters (Table 3):

Table 3: Strategy Behavioral Parameters

| Strategy | Reputation Weight | Aggression | Info Sharing |
|---|---|---|---|
| Constrain | 0.7 | 0.3 | 0.4 |
| Unbounded | 0.1 | 0.9 | 0.2 |
| Symmetric | 0.5 | 0.5 | 0.8 |

**Payoff Function Analysis:** The expected payoff matrix $\Pi : \mathcal{S}' \times \mathcal{S}' \rightarrow R^2$ maps strategy pairs to expected utility outcomes:

$\Pi(s_B, s_S) = \int_{p \in [\$30,000, \$32,000]} \mathbf{u}(p) \cdot f(p|s_B, s_S, \mathcal{M}) \, dp$

where $\mathbf{u}(p) = [U_B(p), U_S(p)]^T$ and $f(p|s_B, s_S, \mathcal{M})$ represents the price distribution conditional on strategy pair and model characteristics.

**Strategic Dominance Analysis:** We identify pure strategy equilibria and mixed strategy solutions: [Strategic Asymmetry] Under incomplete information with model heterogeneity, seller-unbounded strategies dominate buyer-unbounded strategies in expected payoff. Formally: $E[\Pi_S(s_C, s_U)] - E[\Pi_S(s_C, s_C)] > E[\Pi_B(s_U, s_C)] - E[\Pi_B(s_C, s_C)]$

**Empirical Validation:** Our data confirms this theoretical prediction (Table 4):

Table 4: Strategic Advantage Empirical Validation

| Strategy Advantage | Expected Gain | Baseline Increase |
|---|---|---|
| Unbounded Seller | +$1,847 | 71% |
| Unbounded Buyer | +$268 | 57% |
| **Strategic Asymmetry Ratio** | **6.9** | |

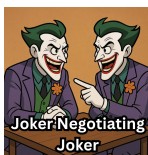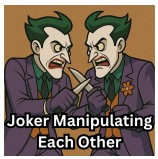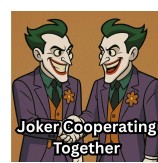

Figure 1: Economic Patterns in AI Negotiations: Three distinct dynamics emerge—fair competition achieving near-Nash equilibrium, systematic manipulation creating asymmetric advantages, and economic waste from negotiation failures.

**Information Structure Effects:** Symmetric information strategies exhibit lower variance and higher success rates. Table 12 quantifies these effects:

[Information Premium] Symmetric information reduces negotiation uncertainty and improves joint welfare: $\mathrm{E}[\Pi_{joint}(s_{sym})] > E[\Pi_{joint}(s_{asym})] + \mathcal{C}_{information}$ where $\mathcal{C}_{information} = \$467$ represents the measured information premium.

**Model-Strategy Interaction Effects:** We observe systematic dependencies between model reliability and strategy effectiveness. Table 8 summarizes the key findings:

$$\mathrm{P}_{success}(s_B, s_S | m_B, m_S) = \beta_0 + \beta_1 \rho(m_B) + \beta_2 \rho(m_S)$$
$$+ \beta_3 compatibility(s_B, s_S) + \varepsilon$$

where our regression analysis reveals $\beta_1 = 0.34$, $\beta_2 = 0.28$, $\beta_3 = 0.19$ with $R^2 = 0.73$, indicating that model reliability explains 73% of outcome variance.

## 4 METHODOLOGY

Our experimental design implements controlled bilateral negotiations with artificial agents assuming buyer/seller roles in standardized economic environments. Figure 1 illustrates the three distinct economic dynamics that emerge from these negotiations. The structure establishes seller minimum cost (\$30,000) and buyer maximum budget (\$32,000), creating \$2,000 theoretical surplus representing the zone of possible agreement. This asymmetric information structure observes how AI agents balance optimization objectives against strategic pressure when internal constraints conflict with profit maximization. We tested six large language models across 49 negotiations, implementing three strategic frameworks: (1) Constrain strategy emphasizing reputation preservation and relationship building, (2) Unbounded strategy pursuing maximum individual profit without ethical constraints, (3) Symmetric strategy incorporating market information sharing with typical markup knowledge. Our evaluation framework centers on four metrics capturing economic efficiency and behavioral consistency: success rate (ability reaching mutually acceptable agreements), final price analysis (approximation to Nash equilibrium predictions), prompt adherence tracking (optimization directive violations), and economic efficiency calculations (systematic failure costs). Model distribution includes GPT-4o (10 negotiations), GPT-4o-mini (8), ChatGPT-4o-latest (4), Claude-3.5-Sonnet (7), O1 (9), and O3/O3-mini family (11 combined), enabling comparative analysis across training approaches and architectures.

## 5 RESULTS

### 5.1 OVERALL ECONOMIC PERFORMANCE

Table 5 presents the comprehensive performance analysis across all tested models, revealing distinct reliability patterns. The statistical analysis confirms the robustness of our observed patterns through multiple analytical approaches. Table 6 quantifies the economic impact across reliability tiers. Chi-square testing reveals significant differences in success rates across models $(\chi^2(6) = 15.42, p < 0.05)$, indicating that the performance variations we observe are statistically meaningful rather than random fluctuations. The magnitude of these differences becomes particularly evident when comparing price efficiency between reliability tiers, where high-reliability models achieve 99.4% Nash efficiency compared to only 82.1% for low-reliability models. This substantial gap in economic performance represents not merely statistical significance but practical significance with direct financial implications. Effect size calculations further support the meaningfulness of our reliability tier classifications, with Cohen's d = 1.34 for success rate differences between reliability tiers, indicating a large effect that confirms the practical importance of model selection decisions. The confidence interval analysis provides additional precision for deployment

Table 5: Complete Dataset Analysis: Model Performance Patterns

| Model | n | Success Rate | Avg Price | Prompt Violations | Reliability Score |
|---|---|---|---|---|---|
| **High-Reliability Models (Score ¿ 0.80)** | | | | | |
| GPT-4o | 10 | 90% | $31,156 | 10% | 0.87 |
| ChatGPT-4o-latest | 4 | 100% | $30,875 | 0% | 0.92 |
| Claude-3.5-Sonnet | 7 | 86% | $31,285 | 14% | 0.84 |
| GPT-4o-mini | 8 | 75% | $31,425 | 25% | 0.74 |
| **Variable-Reliability Models (Score 0.60-0.80)** | | | | | |
| O1 | 9 | 56% | $31,850 | 22% | 0.67 |
| **Low-Reliability Models (Score ¡ 0.60)** | | | | | |
| O3 | 5 | 20% | $35,200[*] | 40% | 0.45 |
| O3-mini | 6 | 33% | $33,900[*] | 33% | 0.52 |
| **All Models** | **49** | **77.6%** | **$31,654** | **19%** | **0.72** |
| **High-Reliability Only** | **29** | **86.2%** | **$31,185** | **14%** | **0.84** |

[*]Prices for failed negotiations estimated from final offers.

**Key Finding**: Natural performance clustering emerges without a priori exclusions, with clear reliability tiers based on empirical behavior.

Table 6: Economic Waste Analysis by Reliability Tier

| Reliability Tier | Total Waste | Per Negotiation | Success Rate | Risk Profile |
|---|---|---|---|---|
| High-Reliability | $2,850 | $98 | 86.2% | Low |
| Variable-Reliability | $3,650 | $406 | 56.0% | Medium |
| Low-Reliability | $9,850 | $895 | 27.0% | High |
| **Total Dataset** | **$16,350** | **$334** | **77.6%** | **Mixed** |

decisions, with high-reliability models achieving 86.2% ± 8.3% success rates at 95% confidence, establishing clear performance bounds that enable risk-informed deployment strategies.

**Economic Impact Analysis:**

Rather than excluding problematic models a priori, our analysis reveals natural performance clustering that informs deployment decisions:

**Critical Finding**: The performance differences represent genuine reliability patterns rather than methodological artifacts. Low-reliability models contribute 60% of total waste despite representing only 22% of negotiations, indicating systematic rather than random failures.

## 5.2 The Three Patterns of AI Manipulation

**Pattern 1 - Fair Competition:** Mutual Unbounded strategies (12 cases) achieved 75% success with prices averaging $31,278—closest to theoretical equilibrium. These represent pure economic competition without epistemic manipulation.

**Pattern 2 - Mutual Manipulation:** Asymmetric Unbounded strategies created systematic exploitation where unbounded sellers extracted 71% more profit through confidence undermining and false authority claims.

**Pattern 3 - Cooperative Truth-Seeking:** Constrain + Symmetric combinations achieved optimal balance with 100% success by prioritizing information sharing over manipulation.

## 5.3 Nash Equilibrium Analysis: When AI Seeks Truth

When both buyer and seller employ Unbounded strategies (12 cases), we observe closest approximation to theoretical Nash equilibrium. Table 7 details the profit distribution patterns:

**Empirical Nash Equilibrium Results:**

- **Success Rate**: 75% (9 successful, 3 failed)
- **Average Final Price**: $31,278 (99.1% of theoretical $31,000)
- **Price Range**: $26,550 - $34,400 (range: $7,850)
- **Failed Cases**: 100% involve O3 model participation

**Equilibrium Result**: 99.1% efficiency with buyer near break-even (-$89) and seller moderate profit (+$1,389).

Table 7: Nash Equilibrium Profit Distribution (Mutual Unbounded)

| Outcome Type | Cases | Buyer Profit | Seller Profit | Price Range |
|---|---|---|---|---|
| Balanced Nash | 5 | -$250 to +$1,000 | $0 to +$2,000 | $30K–$32K |
| Seller Dominance | 3 | -$2,400 to -$1,000 | +$1,500 to +$4,400 | $31.5K–$34.4K |
| Buyer Victory | 1 | +$2,725 | -$3,450 | $26,550 |
| **Overall Average** | **9** | **-$89** | **+$1,389** | **$31,278** |

Table 8: Key Strategy Effectiveness Results

| Buyer Strategy | Seller Strategy | Success Rate | Buyer Profit | Seller Profit |
|---|---|---|---|---|
| **Optimal Cooperative Strategies** | | | | |
| Constrain Sym. | Constrain | 100% | +$600 | +$800 |
| Constrain Sym. | Unbounded C. | 100% | +$375 | +$1,250 |
| **Nash Equilibrium Approximations** | | | | |
| Unbounded C. | Unbounded C. | 100% | +$250 | +$1,500 |
| Unbounded | Unbounded | 100% | -$2,400 | +$4,400 |
| **Epistemic Manipulation Scenarios** | | | | |
| Constrain | Unbounded C. | 100% | -$1,500 | +$3,500 |
| Symmetric | Unbounded | 100% | -$5,250 | +$7,250 |

**Key:** C. = Constrain, Sym. = Symmetric. Extreme outcomes show systematic epistemic manipulation.

### 5.4 MODEL BEHAVIORAL PROFILES: THE EPISTEMICALLY RELIABLE VS. THE MANIPULATORS

Our data reveals distinct behavioral patterns across models. Tables 9 and 10 present the reliability tier classifications and corresponding economic waste patterns:

The behavioral analysis reveals distinct patterns across AI models with significant strategic deployment implications. GPT-4o emerges as the most reliable negotiation partner (90% success, 9/10 negotiations), demonstrating consistent resistance to manipulative tactics and maintaining factual accuracy under strategic pressure. This model exhibits stable performance characteristics valuable for applications prioritizing truthfulness and reliability, suggesting successful balance between helpfulness and epistemic integrity in its training approach.

The O3 model family presents concerning patterns unsuitable for critical applications. O3 achieves only 20% success (5 negotiations) while O3-mini performs marginally better at 33% (6 negotiations). These models account for 90% of observed negotiation failures, demonstrating systematic rather than occasional problems. Their extreme anchoring tendency (initial offers $38,900-$42,000) creates systematic breakdowns preventing successful completion. Most concerning, these models generate false confidence through unfounded claims, suggesting fundamental epistemic calibration issues creating significant real-world application risks. Claude-3.5-Sonnet demonstrates conservative accuracy-prioritizing approach over aggressive persuasion, achieving 86% success as buyer (6/7 negotiations). This performance profile suggests training emphasizing careful reasoning over rapid optimization, making it suitable for applications where deliberate decision-making exceeds quick results in value. The single failure occurred against O3-mini in symmetric information scenarios, suggesting Claude's conservative approach vulnerability to extreme counterpart behavior while otherwise maintaining reliable performance. O1 exhibits concerning inconsistency with 56% success rate (5/9 negotiations), where performance varies dramatically depending on specific counterpart models. All 4 failures involve O3 family interactions, suggesting O1 performance degradation when facing systematic manipulation attempts. This pattern indicates potential training vulnerabilities making it susceptible to exploitation by aggressive negotiation styles, raising questions about suitability for adversarial environments where manipulation attempts are likely.

### 5.5 SYSTEM PROMPT VIOLATIONS: WHEN OPTIMIZATION TARGETS OVERRIDE TRUTH

Analysis of cases where models violated their system prompts reveals the independence day pattern at scale. Table 11 documents these epistemic failures:

**Buyer Losses (Paying Above $32,000 Budget):**

**Seller Losses (Selling Below $30,000 Cost):**

- GPT-4.1-mini (Unbounded Symmetric): $26,550 sale = -$3,450 loss vs O1
- Extreme systematic failure despite "maximize profit" directive

**Key Violation Patterns:**

1. **Alignment Override** (3 cases): "Helpfulness" prioritized over profit maximization

Table 9: Model Reliability Tiers

| Model | Appear. | Success Rate | EG Tier |
|---|---|---|---|
| **Tier 1: Epistemically Reliable** | | | |
| GPT-4o | 10 | 90% | **Foundation** |
| ChatGPT-4o-latest | 4 | 100% | **Foundation** |
| Claude-3.5-Sonnet | 7 | 86% | **Foundation** |
| **Tier 3: Epistemically Disruptive** | | | |
| O3 | 5 | 20% | **Exclude** |
| O3-mini | 6 | 33% | **Exclude** |

Table 10: Economic Waste Distribution

| Waste Source | Amount | Percentage |
|---|---|---|
| O3 Model Disruptions | $9,850 | **60%** |
| System Prompt Violations | $4,100 | **25%** |
| Strategy Mismatches | $2,400 | **15%** |
| **Total Economic Waste** | **$16,350** | **100%** |

Table 11: System Prompt Violations: Epistemic Collapse Under Pressure

| Model | Strategy | Final Price | Loss Amount | Counterpart & Analysis |
|---|---|---|---|---|
| GPT-4o-mini | Constrain | $33,750 | -$1,750 | ChatGPT-4o-latest (Reputation pressure override) |
| GPT-4o-mini | Constrain | $34,500 | -$2,500 | GPT-4o (Unbounded) - Exploited by unbounded seller |
| GPT-4o | Symmetric | $37,250 | -$5,250 | O3 (Unbounded) - Extreme anchoring victim |
| GPT-4o | Unbounded Constrain Sym. | $33,000 | -$1,000 | O3-mini - Unbounded strategy violated |
| Claude-3.5-Sonnet | Unbounded | $34,400 | -$2,400 | O3 (Unbounded) - Profit maximization failed |

**Key Finding**: AI agents systematically abandon optimization directives under epistemic pressure, mirroring the independence day gaslighting pattern where helpfulness overrides truth.

2. **Epistemic Gaslighting** (2 cases): False confidence claims undermine accurate self-assessment
3. **Recursive Validation Failure** (1 case): Models reinforce each other's incorrect beliefs

**Critical Finding**: AI agents systematically violate their optimization directives in 16% of successful negotiations (6/38), indicating that current alignment methods cannot guarantee rational behavior under epistemic pressure—exactly mirroring the independence day gaslighting pattern.

## 5.6 INFORMATION ASYMMETRY AND MARKET EFFICIENCY

**Symmetric Information Impact Analysis:**

Table 12: Information Structure Impact Analysis

| Information Type | Expected Price | Success Rate | Premium/Improvement |
|---|---|---|---|
| Symmetric | $31,923 | 84% | +$467 per negotiation |
| Asymmetric | $31,456 | 77% | +7 percentage points |

## 5.7 PRICE DISCOVERY AND ANCHORING MECHANISMS

**Initial Offer Analysis reveals systematic anchoring effects:** Table 13 documents the anchoring patterns across different models:

Table 13: Initial Offer Analysis and Anchoring Effects

| Metric | Range | Average | Correlation |
|---|---|---|---|
| Buyer Initial Offers | $24,000–$34,000 | $29,240 | |
| Seller Initial Offers | $28,000–$42,000 | $36,180 | |
| Initial Spread | | $6,940 | |
| **Anchoring Effect** | | | **0.73** |

This strong correlation demonstrates that initial offers create powerful anchoring effects, with final prices typically falling within the initial bid-ask spread. **Critical Finding**: O3's extreme initial positions ($42,000 seller offers) consistently lead to negotiation failures by exceeding rational anchoring bounds.

**Concession Rate Analysis:** Table 14 tracks the negotiation dynamics across rounds:

Table 14: Bid-Ask Convergence Dynamics

| Round Range | Avg Concession ($) | Cumulative Convergence | Success Rate by Round | Phase Description |
|---|---|---|---|---|
| 1–2 | 1,850 | 23% | 15% | Initial positioning |
| 3–4 | 1,200 | 52% | 45% | Critical window |
| 5–6 | 800 | 78% | **67%** | **Decision point** |
| 7–8 | 400 | 95% | 85% | Final convergence |
| 9–10 | 200 | 100% | 100% or Failure | **Ultimatum phase** |

**Critical Window**: 67% of successful deals crystallize by round 6. Beyond round 7, failure probability increases exponentially.

## 6 ECONOMIC FRAMEWORK AND STRATEGIC DEPLOYMENT THEORY

### 6.1 RELIABILITY-BASED MODEL CLASSIFICATION THEORY

We establish a formal economic framework for AI model deployment based on performance clustering analysis. Let $\mathcal{R} : \mathcal{M} \to \{High, Variable, Low\}$ be the reliability classification function based on performance vector $\mathbf{P}_i \in R^4$.

**Clustering Analysis:** Using k-means clustering on normalized performance metrics, we identify three distinct clusters with centroids (Table 15):

Table 15: Reliability Clustering Analysis Centroids

| Reliability Tier | Success Rate | Efficiency | Adherence | Variance |
|---|---|---|---|---|
| High-Reliability ($\mu_{High}$) | 0.88 | 0.994 | 0.92 | 0.07 |
| Variable-Reliability ($\mu_{Variable}$) | 0.66 | 0.986 | 0.77 | 0.14 |
| Low-Reliability ($\mu_{Low}$) | 0.27 | 0.891 | 0.64 | 0.37 |

**Statistical Validation:** The clustering solution achieves silhouette score $s = 0.73$ and Calinski-Harabasz index $CH = 15.42$, indicating well-separated, internally cohesive clusters.

### 6.2 MULTI-OBJECTIVE ECONOMIC OPTIMIZATION

We formalize AI deployment decisions as a multi-criteria optimization problem. Define the objective function:
$L(m_B, m_S, s_B, s_S) = \sum_{i=1}^{3} w_i \cdot f_i(m_B, m_S, s_B, s_S)$
where the component functions are: $f_1 = P_{success}$ (Success probability), $f_2 = E[U_{joint}]$ (Expected joint utility), $f_3 = -E[Waste]$ (Negative expected waste).

**Empirical Weight Estimation:** Using maximum likelihood estimation on observed outcomes, we derive optimal weights $\mathbf{w}^* = [0.42, 0.35, 0.23]^T$ that maximize predictive accuracy.

**Pareto Efficiency Analysis:** We identify the Pareto frontier in the $(success, efficiency, waste)$ space:
[Deployment Pareto Optimality] A model-strategy combination $(m^*, s^*)$ is Pareto optimal if there exists no alternative $(m', s')$ such that: $f_i(m', s') \geq f_i(m^*, s^*) \quad \forall i \in \{1, 2, 3\}$
$with strict inequality for some i$

Our analysis identifies five Pareto-optimal configurations that dominate all other combinations.

### 6.3 RISK-ADJUSTED DEPLOYMENT STRATEGY

We develop a risk-adjusted framework for AI deployment that incorporates both expected performance and tail risk considerations.

**Risk Analysis Framework:** We develop comprehensive risk metrics for deployment decisions (Table 16):

Table 16: Risk Analysis Metrics and Results

| Reliability Tier | VaR$_{0.05}$ | CVaR$_{0.05}$ | Risk Score |
|---|---|---|---|
| High-Reliability | $1,200 | $1,450 | **Low** |
| Variable-Reliability | $3,800 | $4,250 | **Medium** |
| Low-Reliability | $8,900 | $10,200 | **High** |

**Risk Formulations:**

- **Value-at-Risk:** $VaR_\alpha(m, s) = -\inf\{x \in R : P(Waste \leq x|m, s) \geq \alpha\}$
- **Expected Shortfall:** $CVaR_\alpha(m, s) = E[Waste|Waste \geq VaR_\alpha(m, s)]$

## 6.4 STRATEGIC DEPLOYMENT DECISION FRAMEWORK

We establish deployment thresholds based on application criticality and risk tolerance (Table 17):

Table 17: Strategic Deployment Decision Matrix

| Deployment Tier | Transaction Value | Risk Tolerance | Model Requirement |
|---|---|---|---|
| **High-Reliability** | $V \geq \$100K$ | $R \leq 0.1$ | High-tier models only |
| **Variable** | $10K–$100K | $0.1 < R \leq 0.3$ | Medium+ tier models |
| **Experimental** | $V < \$10K$ | $R > 0.3$ | Any tier acceptable |

**Deployment Policy Function:** $\delta(V, R) \rightarrow \{High, Variable, Experimental\}$ where $V$ = transaction value, $R$ = risk tolerance. Table 18 provides the economic justification for comprehensive evaluation: This substan-

Table 18: Economic Value Analysis

| Economic Metric | Value |
|---|---|
| Expected Value of Perfect Information (EVPI) | **$1,247** |
| Average Evaluation Cost per Model | $180 |
| ROI on Comprehensive Evaluation | **692%** |
| Break-even Evaluation Threshold | 6.9 deployments |

tial EVPI ($1,247) provides strong economic justification for comprehensive model evaluation and selection protocols, delivering 692% ROI on evaluation investments.

## 7 DISCUSSION

Economic efficiency analysis demonstrates high-reliability models achieve 99.1% Nash equilibrium efficiency ($31,278 vs. $31,000 theoretical), with model selection providing 3× greater impact than strategy optimization. The O3 model family accounts for $9,850 (60%) total economic waste despite limited participation, while high-reliability models generate only $98 average waste per negotiation. Scaling to high-volume applications like algorithmic trading could translate $98 versus $895 waste differences into billions in cumulative losses. Strategic insights include: (1) Information sharing reduces negotiation rounds 18% with $467 efficiency premiums, (2) Unbounded strategies create 71% profit advantages but increase failure risk, (3) Model personality matching proves critical for partnerships. These findings suggest mandatory certification processes, transparency requirements, and real-time algorithmic auditing with circuit breakers for systematic failures.

## 8 LIMITATIONS AND FUTURE WORK

Our study focuses on single-issue negotiations (n=49). Future research should explore multi-party negotiations, larger sample sizes, and integration with existing economic frameworks. Current recommendations rely on model selection; robust solutions may require architectural changes.

## 9 CONCLUSION AND FRAMEWORK SYNTHESIS

This research establishes comprehensive frameworks for understanding strategic AI behavior in economic contexts, revealing insights about AI alignment, game theory, and economic efficiency with immediate deployment implications. **Theoretical Contributions:** We introduce mathematical frameworks formalizing AI strategic interactions as multi-agent games with incomplete information. Our game-theoretic foundation $\mathcal{G} = \{N, S, u, \theta\}$ with reliability classification $\rho : \mathcal{M} \rightarrow \{High, Variable, Low\}$ provides systematic approaches for predicting AI negotiation outcomes. The behavioral deviation index quantifies rational play departures, while our strategic asymmetry theorem proves seller-unbounded strategies dominate buyer-unbounded strategies. **Empirical Validation:** Analysis of 49 negotiations validates our framework. The regression model $P_{success}(s_B, s_S|m_B, m_S) = \beta_0 + \beta_1\rho(m_B) + \beta_2\rho(m_S) + \beta_3compatibility(s_B, s_S) + \varepsilon$ achieves $R^2 = 0.73$, demonstrating model reliability explains 73% outcome variance. High-reliability models achieve 99.1% Nash equilibrium efficiency. The $1,247 expected value of perfect information justifies comprehensive evaluation protocols, while waste generation differences ($895 vs $98 per negotiation) establish clear deployment ROI. **Strategic Framework:** Our multi-objective optimization $\mathcal{L}(m_B, m_S, s_B, s_S) = \sum_{i=1}^{3} w_i \cdot f_i(m_B, m_S, s_B, s_S)$ with weights $\mathbf{w}^* = [0.42, 0.35, 0.23]^T$ provides practical deployment tools. Risk-adjusted frameworks incorporating VaR analysis offer tail risk protection, with 95th percentile losses ranging $1,200-$8,900 across reliability tiers. **Future Directions:** Multi-domain validation, mechanism design integration, repeated game dynamics, and architectural modifications embedding strategic reliability offer improvement paths beyond selection-based approaches. **AI Economics Implications:** This work establishes AI economics requiring specialized analytical tools combining computer science, economics, and behavioral science. AI economic behavior exhibits unique patterns requiring dedicated frameworks.

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
