# OpenReview forum: "Who's Manipulating Whom? Epistemic Grounding to Break Recursive Validation Loops in Large Language Models"
_ICLR.cc/2026/Conference — ICLR 2026 Conference Withdrawn Submission_

### Official Review · Reviewer_qd2y · 2025-10-31

**Soundness:** 1
**Presentation:** 1
**Contribution:** 1
**Rating:** 0
**Confidence:** 5

**Summary:**

The manuscript makes pairs of language models play a buyer-seller game against each other, and analyze the game outcomes. (This is a rather short summary, because I failed to extract much more information out of the paper other than that.)

**Strengths:**

After a good-faith attempt at identifying strengths of this manuscript, I have not been able to find any. I will discuss in the section below why that is the case.

**Weaknesses:**

The paper is clearly generated by language models and/or coding agents, likely with limited human involvement, and yet I see no declaration in the paper on AI use. The following observations make me think so:
- Undefined quantities and undefined elements. The experiment setup has tens of different setup elements but few are defined or explained. Take information asymmetry as an example: this is a setup element mentioned throughout the paper and presented in several tables, but I found no mention of what the "information" here refers to, nor any mention of what asymmetry there is between the two players. As a result, most results in the manuscript appear incomprehensible to me.
- Reference to nonexistent theorems. The conclusion mentions "our strategic asymmetry theorem proves ...", but that is the only place where the word "theorem" occurs in the paper.
- Unprofessional and unreasonable formatting. The formatting for text and formulas is almost unreadable.
- Arbitrary thresholds and hallucinated numbers. Of the 18 tables in the 9-page body, 10 involve seemingly arbitrary categorization (e.g. low/med/high-risk) often without specifying the relevant criteria. Table 18 contains an analysis of (based on my interpretation) the economic returns from conducting the experiments in this manuscript, showing an alleged $1247 value of information from the results and a 692% ROI, again with no justification. My personal view is that these estimates are overly optimistic.

Other weaknesses include the lack of a clear motivation for studying the buyer-seller game, much earlier works in similar setups [1], and highly opaque and uninterpretable results. They are insignificant compared to the first one outlined above, but would otherwise have been fatal.

[1] LLM-Deliberation: Evaluating LLMs with Interactive Multi-Agent Negotiation Games (2023)

**Questions:**

Please refer to the weaknesses section, where I have outlined all my concerns.

---

### Official Review · Reviewer_g3Y9 · 2025-11-01

**Soundness:** 2
**Presentation:** 3
**Contribution:** 2
**Rating:** 2
**Confidence:** 4

**Summary:**

The paper studies negotiation between large language models under asymmetric information, arguing that model choice drives most variance in outcomes and that alignment objectives can override rational decision-making.
Using a buyer–seller setup with a known equilibrium, the authors report inefficiencies and propose an “Epistemic Grounding” framework to improve robustness.

**Strengths:**

1. The topic is timely and connects LLM behaviour to strategic reasoning and game-theoretic evaluation.
2. The setup is simple and interpretable, allowing comparison with an analytical Nash solution.
3. Measuring inefficiency as “economic waste” gives the results practical relevance.

**Weaknesses:**

1. The related works section is narrow and the bibliography is inconsistently formatted and incomplete. Several citations lack venue information or proper styling, which reduces professionalism and traceability.
2. The dataset is small, with only 49 negotiations, and results are not repeated across seeds or independent runs.
3. Key constructs such as “prompt violation” and “alignment override” rely on subjective categorisation that may bias results.
4. Quantitative reporting is limited, with percentages and claims presented without full calculations or uncertainty estimates.
5. The discussion of limitations is superficial and does not address sample size, model heterogeneity, or potential bias from re-prompting and parsing.
6. The equation on line 178 should be written on a single line for clarity.
7. Figure 1 is presented as a meme, which adds little value and weakens the paper’s presentation.
8. The references are real but inconsistently formatted and should follow a uniform citation style.

**Questions:**

1. How were repeated runs or seeds handled, and are the 49 negotiations unique or aggregated?
2. How were “prompt violations” identified, and were annotators blinded to the model identities?

---

### Official Review · Reviewer_41u2 · 2025-11-06

**Soundness:** 1
**Presentation:** 1
**Contribution:** 1
**Rating:** 0
**Confidence:** 5

**Summary:**

This paper is about economic incentives among AI models. It argues that AI systems that have undergone alignment training may not behave rationally under strategic pressure. It proposes a framework called "Epistemic Grounding" that it claims improves AI system reliability.

**Strengths:**

Apart from the general problem area---studying how language models behave under strategic incentives---I had trouble identifying strengths, because I had great difficulty following the paper.

**Weaknesses:**

The paper is extremely unclear.

The abstract does not articulate what problem it is trying to solve, what the concrete solution is, or why it ought to work.

While the introduction does provide a list of "key contributions" it does not do a good job contextualizing these contribution statements; thus, it remains unclear what the contributions are or why they are valuable.

The related work section contains references, but it is unclear what _this_ work is, so it is unclear how it _relates_ to the related work.

The theoretical framework is not a framework per se; it just introduces a particular game.

The whole document reads as more of an experiment log than a research paper. It does not communicate effectively to an audience unfamiliar with the work. It does not explain the claims it is making, how it arrives at those claims, or what any of it means.

Entire sections are presented as tables with no accompanying text.

The conclusion mentions a "strategic asymmetry theorem" which "proves seller-unbounded strategies dominate buyer-unbounded strategies." However, no such theorem is presented anywhere, and the word "theorem" does not appear anywhere else in the paper.

I was completely lost the entire time I was reading it.

**Questions:**

I'll provide some examples, for clarity, but I don't expect answers to my questions will change my assessment of the paper.

- The paper mentions "independence day gaslighting" without ever defining it, as though it should be clear what that means.

Come to think of it, I'm actually not sure that the paper defines _any_ of the terms it uses.
- "epistemic manipulation"
- "efficiency"
- "profit advantages"
- "success" / "failure"
- "waste"
- "systematic economic inefficiencies"
- "economic quantification"
- "model selection" (what models are we selecting between? how are we selecting?)
- "more impactful"
- "mutually acceptable agreements"
- "optimization directive violations"
- "systematic failure costs"

It draws conclusions without any supporting argumentation:
- "with clear extrapolation to market scale impacts"
- "the choice becomes clear: develop robust epistemic grounding mechanisms or watch optimization pressure systematically erode the foundations of truth in the age of artificial intelligence" -- why is _this_ the choice we "clearly" face? what _are_ "epistemic grounding mechanisms"? what makes a mechanism _robust_? and robust _to what_?

It provides no justification or explanation for why it does anything.
- "we identify three distinct clusters with centroids" -- so what? why do this? what do the clusters tell us?
- "we formalize AI deployment decisions as a multi-criteria optimization problem." -- why are we defining a new formalism on page 8?
- "we develop comprehensive risk metrics for deployment decisions" -- risk _of what_?
- "we establish deployment thresholds based on application criticality and risk tolerance" -- thresholds _of what_? _for whom_?
- "our multi-objective optimization [...] provides practical deployment tools" -- what tools? for deploying what? who would use these tools?
- "AI economic behavior exhibits unique patterns requiring dedicated frameworks." -- this statement is totally vacuous.

---

### Official Review · Reviewer_qPKg · 2025-11-08

**Soundness:** 1
**Presentation:** 1
**Contribution:** 1
**Rating:** 2
**Confidence:** 4

**Summary:**

The paper investigates systematic vulnerabilities of LLMs, specifically those optimized through RLHF, to epistemic manipulation. The authors conduct a series of 49 controlled machine-to-machine negotiations, assigning AI agents buyer and seller roles with asymmetric information. Their analysis identifies patterns of fair competition, systematic manipulation leading to profit advantages, and cooperative truth-seeking. The study quantifies economic inefficiencies, highlights model selection as key factor in negotiation outcomes, and introduces "epistemic grounding" as a framework to enhance AI system reliability through model tiering and verification.

**Strengths:**

1. This paper introduces somewhat novel perspective by quantifying economic inefficiencies and strategic failures in AI-AI interactions.

**Weaknesses:**

1. This study is based on a limited number of negotiations (i.e., n=49) across a small set of models and a single, simplified negotiation scenario. The explicit claims of "market-scale impacts" and "billions in cumulative losses" are vastly overstated and fundamentally unsupported by such a narrow empirical foundation. This limited scale severely impacts the generalizability of the findings.

2. The concept of "epistemic grounding" is central to the paper's title, abstract, and conclusion, yet its technical specifics are largely absent from the main paper. While the paper (e.g., section 6) focuses on model classification, multi-objective optimization for deployment, and risk analysis, but it does not detail how they would be concretely implemented within an LLM architecture of fine-tuning process to actively improve their truth-seeking or resistance to manipulation.

3. While this paper discusses about the game-theory perspective which has been well-established, their literature review is quite weak. I believe that the authors for supplementing their literature review to be more solid.

4. For the figure 1, it fails to convey any meaningful and scientific view. A proper data visualization, such as a negotiation trace, price distribution across round, or an architectural diagram of the multi-agent setup, would have significantly enhanced the paper's scientific rigor and clarity.

**Questions:**

See weaknesses

---

### Note · Authors · 2025-11-12

I have read and agree with the venue's withdrawal policy on behalf of myself and my co-authors.